# Membrane-localized expression, production and assembly of *Vibrio parahaemolyticus* T3SS2 provides evidence for transertion

Karan Gautam Kaval [1,2], Suneeta Chimalapati[3], Sara D. Siegel [4], Nalleli Garcia[5], Jananee Jaishankar[1], Ankur B. Dalia [6] & Kim Orth [1,2,7]

It has been proposed that bacterial membrane proteins may be synthesized and inserted into the membrane by a process known as transertion, which involves membrane association of their encoding genes, followed by coupled transcription, translation and membrane insertion. Here, we provide evidence supporting that the pathogen *Vibrio parahaemolyticus* uses transertion to assemble its type III secretion system (T3SS2), to inject virulence factors into host cells. We propose a two-step transertion process where the membrane-bound co-component receptor (VtrA/VtrC) is first activated by bile acids, leading to membrane association and expression of its target gene, *vtrB*, located in the T3SS2 pathogenicity island. VtrB, the transmembrane transcriptional activator of T3SS2, then induces the localized expression and membrane assembly of the T3SS2 structural components and its effectors. We hypothesize that the proposed transertion process may be used by other enteric bacteria for efficient assembly of membrane-bound molecular complexes in response to extracellular signals.

Large molecular machines such as the Type III Secretion System (T3SS) are conserved amongst a multitude of Gram-negative enteric pathogenic bacteria. Owing to the sheer complexity of this macromolecular injectisome, its expression as well as assembly, must be tightly regulated to correspond to specific cues within the host environment to ensure correct localization during the appropriate stage of infection[1]. For this very purpose, bacteria have evolved sophisticated signaling mechanisms to sense and adapt to changing host conditions primarily using one- and two-component systems[2,3]. While two-component signaling systems employ a separate transmembrane kinase for signal sensing and a separate regulator that diffuses through the cytoplasm to reach its regulatory target, one-component systems largely utilize a single cytoplasmic protein bearing both the sensing and regulatory domains, thereby serving both purposes[2,3]. Although the majority of

one-component regulators/transcription factors are soluble, a minority of one-component regulators/transcription factors are bitopic, characterized by the presence of a ligand-binding periplasmic sensor domain, and a cytoplasmic helix-turn-helix (HTH) DNA-binding domain connected via a single-pass transmembrane region[3]. The ToxR family of receptors, including ToxR, TfoS, and TcpP of *Vibrio cholerae*[4], CadC of *E. coli*[5], and PsaE of *Yersinia*[6] has long been described to belong to this class of transmembrane one-component signal transduction systems. All the receptors in this family possess the characteristic CadC superfamily of DNA-binding HTH domains, their periplasmic sensor domains having diverse subdomain architectures, with some requiring transmembrane coregulators for activity[5,7–11]. A recent study demonstrated that the members of the ToxR family requiring coregulators for ligand-binding and stability, such as

[1]Department of Molecular Biology, University of Texas Southwestern Medical Center, Dallas, TX 75390, USA. [2]Howard Hughes Medical Institute, University of Texas Southwestern Medical Center, Dallas, TX 75390, USA. [3]Reata Pharmaceuticals, Plano, TX 75204, USA. [4]Biomanufacturing Training and Education Center, North Carolina State University, Raleigh, NC 27606, USA. [5]Department of Microbiology and Cell Science, University of Florida, Gainesville, FL 32611, USA. [6]Department of Biology, Indiana University, Bloomington, IN 47405, USA. [7]Department of Biochemistry, University of Texas Southwestern Medical Center, Dallas, TX 75390, USA. e-mail: kim.orth@utsouthwestern.edu

periplasmic heterodimers ToxR/ToxS of *Vibrio cholera*[7,10] and VtrA/VtrC of *Vibrio parahaemolyticus*[9,12], serve as co-component signal transduction systems rather than as traditional one-component systems[13].

As the target genes regulated by such co-component signal transduction systems encode at least one membrane integral protein[4,9,14], their mode of action has been theorized to first involve the localization of these genes to be proximal to the membrane upon induction[15–17]. Subsequently, the process termed as "transertion", characterized by concurrent transcription, translation, and membrane insertion of biosynthetic products from their membrane-localized target genes, is thought to occur, resulting in membrane assembly of these proteins[15–18].

The syringe-like Type III Secretion System 2 (T3SS2) injectisome of the Gram-negative gastrointestinal pathogen, *Vibrio parahaemolyticus*, is a multiprotein complex consisting of ~19 core structural components, all of which are essential for the functionality of this apparatus[19,20]. The membrane-bound heterodimeric co-component signal transduction system, VtrA/VtrC, activates expression of the genes encoding this apparatus upon induction with bile acids, thereby mediating the production and injection of specialized effector proteins into host cells to manipulate host cellular processes[13,21–23]. Like VtrA/VtrC, VtrB is an inner membrane protein with a cytoplasmic N-terminal HTH DNA-binding domain followed by a transmembrane helix that tethers it to the membrane. It is a master regulator of the T3SS2 that binds the promoters of genes encoding T3SS2 components, facilitating transcriptional activation by VtrA/VtrC[12,22].

In this work, we report that the colocalized expression, production, and membrane assembly of the T3SS2 apparatus in *V. parahaemolyticus* is consistent with the transertion mechanism occurring in two steps mediated by two membrane-embedded transcription factors that work in tandem. Binding of bile acids to the periplasmic domain of the transmembrane VtrA/VtrC complex induces dimerization and activation of its cytoplasmic DNA-binding HTH, allowing it to capture the T3SS2 pathogenicity island at the membrane via the encompassed *vtrB* promoter (Fig. 1a). This membrane capture initiates the first transertion step allowing for the expression, assembly, and activation of the membrane-bound VtrB transcription factor (Fig. 1b). As a master regulator of the T3SS2, VtrB then carries out the second transertion step by binding to its target promoters within the membrane-associated T3SS2 pathogenicity island and inducing transcription resulting in localized production of the T3SS2 components at the membrane. These two transertion steps enable the efficient assembly of the T3SS2 apparatus with its chaperone-bound effectors (Fig. 1c).

## Results

### *vtrB* genomic locus is captured at the inner membrane by VtrA/VtrC upon bile acid-induction

To better understand the effect of bile acid-mediated VtrA/VtrC activation on the membrane proximity of the *vtrB* genomic locus, a locus-tagging and tracking methodology based on a minimal bacterial plasmid-partitioning Par system visualized with super-resolution microscopy was employed[24] (Fig. 2a, b). In the *V. parahaemolyticus* CAB2 strain background[25] (Supplementary Table. 1), we introduced a *parSMT1* sequence from the *E. coli* MT1 plasmid in the intergenic region downstream of the *vtrB* genomic locus. The cognate fluorescently labeled ParBMT1 was expressed under the control of an arabinose-inducible promoter, to tag and track the genomic *vtrB* locus (Fig. 2a). These cells were cultured in the presence or absence of the activating bile salt, taurodeoxycholate (TDC). Localization of the fluorescently labeled *vtrB* loci within the cells (Fig. 2b) was observed and quantified using the super-resolution module with a magnification of ×320 (~100 nm in xy and ~400 nm in z) by measuring their distances from the membrane (Fig. 2c, d). Distribution of normalized distances of the *vtrB* loci from the membrane (p') revealed membrane bias of this

locus in cells grown with TDC as compared to those in cells grown in non-inducing conditions (Fig. 2e). A neutral locus (0.458 Mb on chromosome 2 located outside the T3SS2 pathogenicity island) was monitored simultaneously with the T3SS2 locus using an analogous Par system from the *E. coli* P1 plasmid[24] (Supplementary Fig. 1a, b). The neutral locus displayed no differences in its distribution within cells cultured either with or without TDC (Supplementary Fig. 1c). By contrast, when a copy of the *vtrB* promoter was integrated into the neutral locus, this site exhibited bile acid-dependent membrane bias similar to that of the native *vtrB* genomic locus (Supplementary Fig. 2a, b). Upon tagging and quantifying the *vtrB* genomic locus proximity to the membrane in a Δ*vtrA/vtrC* mutant[25], we observed that the *vtrB* loci displayed mid-cell-bias irrespective of whether the cells were grown in inducing or non-inducing conditions (Supplementary Fig. 2c, d). Collectively, these data demonstrate that upon bile acid-induction, the activated VtrA/VtrC co-component signal transduction system specifically captures the *vtrB* locus at the inner membrane (Fig. 1a).LL

### VtrB and the T3SS2 needle are expressed and assembled at the membrane corresponding to the site of *vtrB* genomic locus capture

To visualize where VtrB-mediated expression of the T3SS2 occurs in cells, VtrB was expressed from its native locus with its periplasmic component tagged with a monomeric superfolding GFP that can efficiently fold and fluoresce within the periplasmic oxidative niche[26]. The membrane proximity of the *vtrB* genomic locus was simultaneously visualized in this background using the *parS*-ParBMT1 locus-tagging system described previously, except with the YGFP tag of ParBMT1 swapped for CFP. Localization of VtrB with the *vtrB* locus at the inner membrane was observed in cells using both super-resolution confocal microscopy (Fig. 3a), as well as widefield fluorescence microscopy (Supplementary Fig. 3a). Correlation analysis comparing the fluorescent signal pixel intensities of VtrB and the *vtrB* locus in cells induced with TDC revealed strong positive correlation between these two labeled molecules, with Pearson's Correlation Coefficients (R) of >0.9 serving as a strong indicator of colocalization (Supplementary Fig. 3b). Quantification of the number of *vtrB* loci, VtrB, and *vtrB* locus-VtrB colocalized puncta uncovered that, while not all *vtrB* loci were associated with a corresponding VtrB signal, there were no significant number of VtrB puncta without an accompanying *vtrB* locus signal (Supplementary Fig. 3c). In the absence of inducing conditions, the *vtrB* locus displayed mid-cell bias with absence of any observable VtrB expression (Fig. 3a). Taken together, these observations indicate that TDC-induction results in the localized expression and assembly of VtrB at the inner membrane via transertion (Fig. 1b).

Of the core structural components of the T3SS2 injectisome, the needle protein is the one that links the membranes of the bacteria and the host cell by forming a filament through the extracellular space, allowing for insertion of the injection apparatus into the host membrane to facilitate transfer of bacterial effectors into the host cytosol[27]. While most of the T3SS2 structural components have been characterized, the identity of the needle filament protein, in addition to four others, remains obscure due to low sequence similarity with other T3SS needle proteins[19,20]. Using sensitive sequence similarity detection methods (HHPRED), we found the T3SS2 protein, Vpa1343, confidently identifies the secretion system needle proteins from *Yersinia* (YcsF, 2P58_B, probability 98.24%), *Salmonella* (PrgI, 2X9C_A, probability 98.19%), *Shigella* (MxiH, 6ZNI_C, probability 98.19%) and *Burkholderia* (BsaL, 2G0U_A, probability 97.24%). Therefore, Vpa1343 is used as an indicator of VtrB-mediated production and assembly of the T3SS2 apparatus.

We chose an established thiol chemistry-based approach to specifically tag and visualize the formation of T3SS2 needles[28]. *V. parahaemolyticus* strains expressing Vpa1343 needle proteins containing a serine to cysteine mutation (S3C, S32C, S62C, S85C, and S91C) were tested for the functionality of their T3SS2 apparatuses in terms of

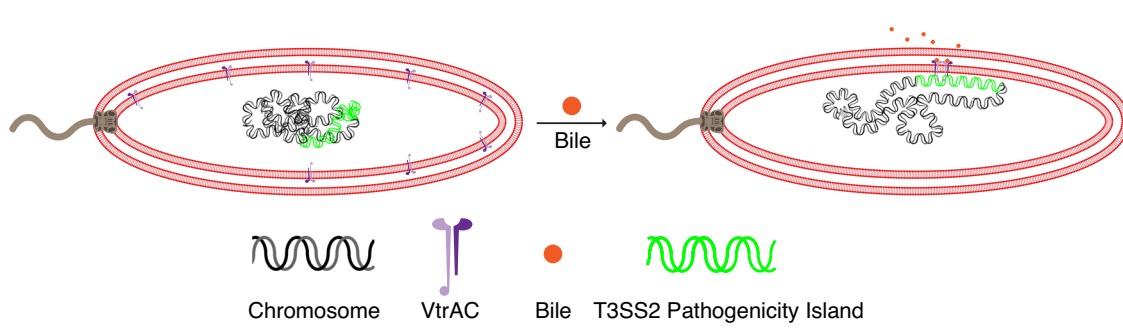

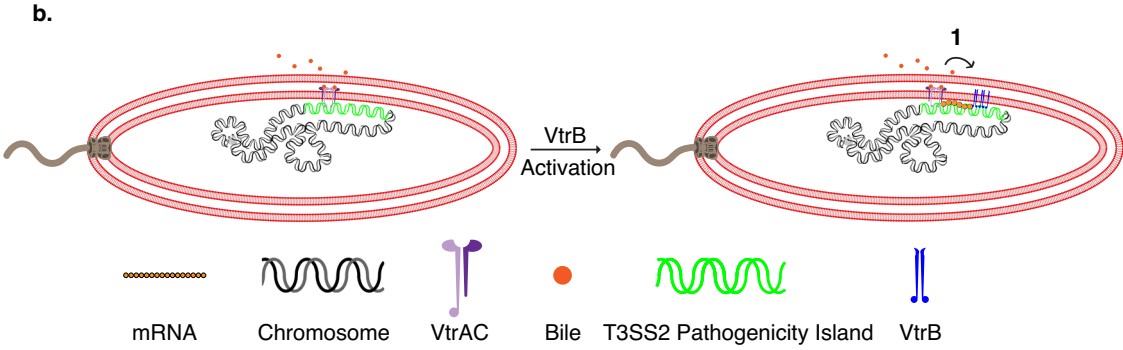

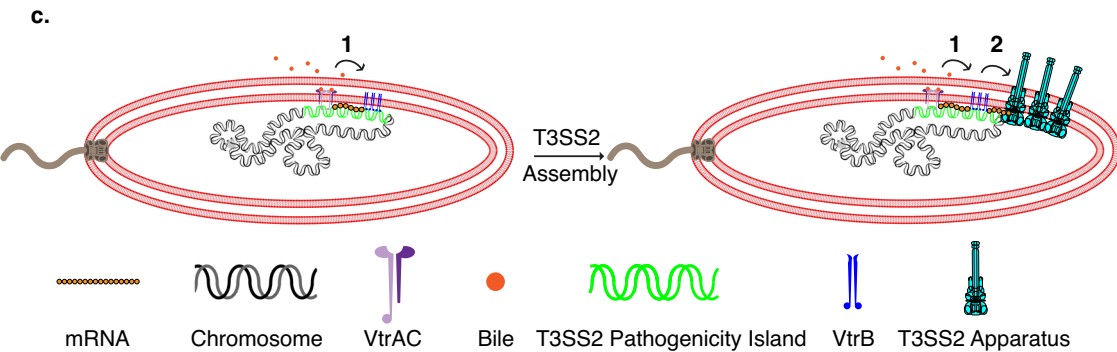

**Fig. 1 | Two-step transertion model for the localized membrane assembly of the T3SS2 apparatus in V. parahaemolyticus.** Illustrations depicting **a** dimerization and activation of VtrA/VtrC upon binding bile acid, which captures the T3SS2 pathogenicity island containing the *vtrB* genomic locus at the inner membrane, **b** first transertion step, starting with transcription initiation of *vtrB* by VtrA/VtrC, and concurrent translation and membrane insertion of the VtrB protein in the immediate vicinity of the membrane-captured *vtrB* genomic locus, **c** second transertion step, starting with transcription initiation of T3SS2 structural and chaperone-associated effector genes by clustered VtrB upon binding to their respective membrane-proximal promoters, and simultaneous translation and membrane assembly of T3SS2 apparatus. **b**, **c** 1 = first transertion step, 2 = second transertion step.

relative cytotoxicity in HeLa cells (Fig. 3b) and effector secretion (Fig. 3c). Collectively, these data indicate that among the tested mutants, Vpa1343S3C was able to produce functional T3SS2 needle filaments and could therefore be tagged with fluorophore-conjugated, thiol-reactive dyes for confocal imaging. The *parSMT1*-ParBMT1 locus-tagging and VtrB-msfGFP constructs were reconstituted in the *V. parahaemolyticus* CAB2 Δ*vpa1343::vpa1343-S3Ck* background, and cells of the resultant strains were grown in either inducing (100 μM TDC) or non-inducing conditions. Using super-resolution and wide-field imaging, the Alexa Fluor™ 647 C₂ Maleimide-stained T3SS2 needles (cyan) were observed to be extracellular, localizing adjacent to the outer membrane near the inner membrane localization of both the *vtrB* genomic locus (Fig. 3d, Supplementary Fig. 5) and the VtrB protein (Fig. 4a, Supplementary Fig. 6a) in inducing conditions for these strains. These needle filaments, however, were not observed in the absence of bile acid-induction (Fig. 3d, Supplementary Fig. 4 and Fig. 4a, upper panels). Maleimide specifically stains surface exposed thiol groups. The selective maleimide-staining of needle subunits, therefore, is observed only on the surface of cells but not within the membrane. This staining results in only minor overlap of needle-associated fluorescence signals with that of the membrane and VtrB (Fig. 4a) or the *vtrB* loci (Fig. 3d). This observation was corroborated by the colocalization analysis of widefield images of the VPKK19 strain, where $0.3 < R < 0.5$ was observed for pixel intensity correlations

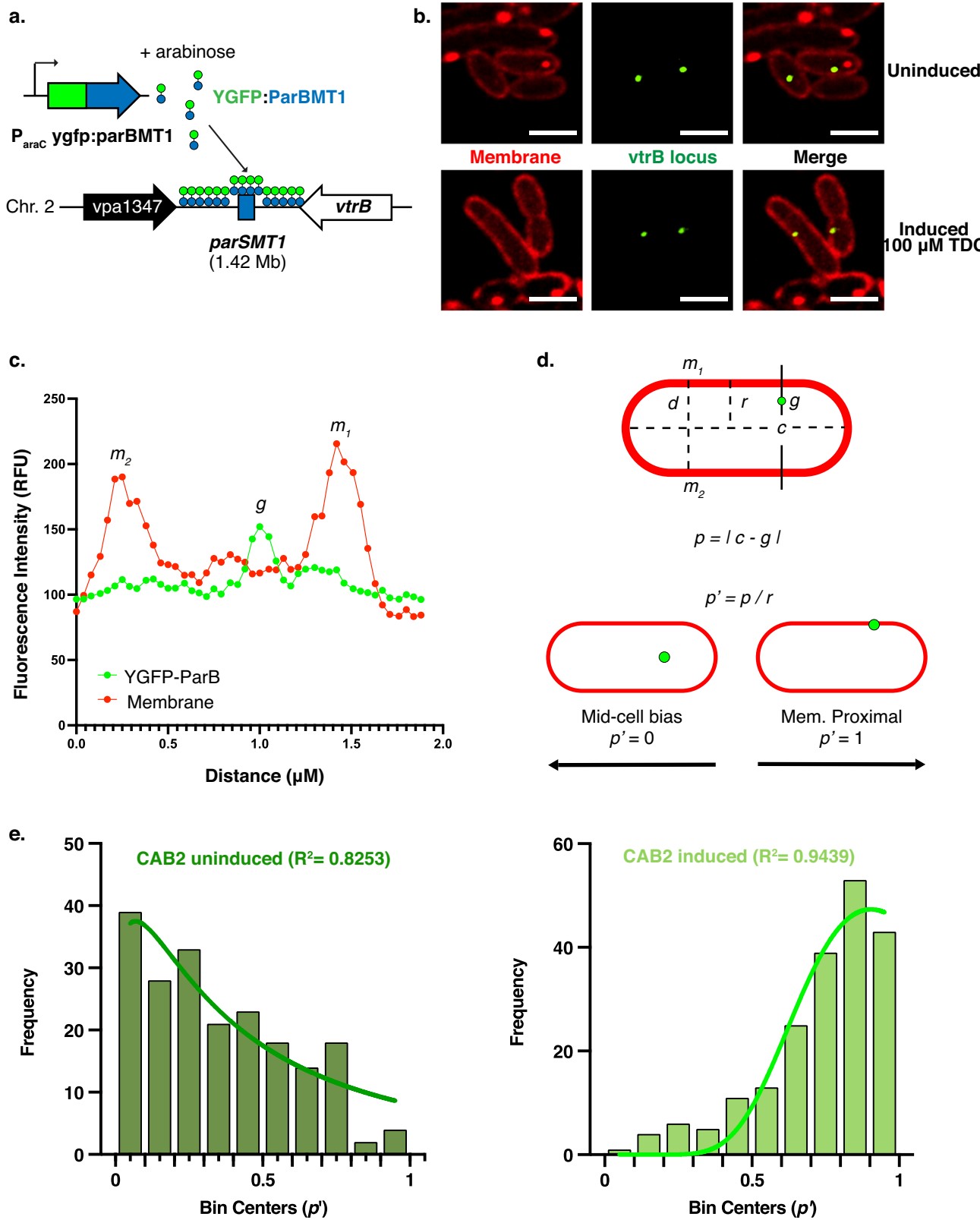

involving the T3SS2 needles and VtrB (Supplementary Fig. 6b). Absolute counts of VtrB, T3SS2 needle, and VtrB + T3SS2 needle localization puncta (Supplementary Fig. 6c) revealed that only ~10% of the quantified cells assembled T3SS2 needles. Importantly, these T3SS2 needle puncta were observed to have accompanying VtrB signals (Supplementary Fig. 6c). Additionally, transmission electron micrographs displayed colocalization of T3SS2 needles and the VtrB protein at the

membrane in bile acid-induced cells, that were stained with their respective immunogold-conjugated antibodies. (Fig. 4b).

## VopV mRNA is locally transcribed at the site of VtrB membrane localization

The primary purpose of the T3SS2 apparatus is to inject specialized effector proteins into the host cell, allowing the bacteria to subvert

**Fig. 2 | Bile salt induction of VtrA/VtrC leads to inner membrane capture of the vtrB locus. a** Illustration depicting *vtrB* genomic locus-tagging using the *parsMT1*-ParBMT1 system in *V. parahaemolyticus*. **b** Confocal micrographs showing the intracellular localization of the *vtrB* genomic loci (green) in relation to the membrane (red) in *V. parahaemolyticus* SS2B5 cells cultured in non-inducing or inducing (100 μM TDC) conditions with 0.02% arabinose supplementation for YGFP-ParBMT1 expression. Membranes were stained with nile red. Scale bar = 2 μm. **c** Fluorescence intensity profile of a *V. parahaemolyticus* SS2B5 cell grown in non-inducing conditions for *vtrB* locus membrane proximity quantification. Intensity peaks $m_1$ and $m_2$ denote lateral membrane fluorescence intensities, while g corresponds to the intensity of the YGFP-ParBMT1 punctum (*vtrB* locus). **d** Illustration describing methodology for the normalization of *vtrB* locus membrane proximity (p') where $g$ = *vtrB* locus (μm), $m_1$ = lateral wall 1 (μm), $m_2$ = lateral wall 2 (μm), were calculated using the fluorescence intensity profile for each YGFP-ParBMT1 punctum. For subsequent calculations, $d$ = diameter (μm), $c$ = cell center (μm), and $r$ = cell radius (μm). **e** Representative Frequency distribution plots with nonlinear regression (lognormal) analyses of normalized *vtrB* loci distances (p') in *V. parahaemolyticus* SS2B5 cells, cultured in non-inducing (left panel) and inducing (100 μM TDC, right panel). $N$ = 200 *vtrB* loci per condition. $R^2$ = goodness of fit of the regression curves. **b, e** Three independent biological repetitions were performed. Source data are provided as a Source Data file.

cellular processes and establish a replicative niche during infection[29]. VopV is such an effector protein that specifically binds to F-actin of the host cells, mediating enterotoxicity[30]. To ascertain localized transcription of a T3SS2 effector, VopV mRNA was visualized using the MS2 system[31], where 24×MS2 bacteriophage RNA hairpins were introduced at its 3′ UTR and was tagged using fluorescently labeled MS2 coat protein (MCP). VtrB and VopV mRNA were observed to colocalize ($R > 0.6$) at the membrane in cells grown under inducing conditions (Supplementary Fig. 7a, b), with no observable VtrB and VopV mRNA signals in the absence of bile acids (Fig. 4c). Absolute counts of VtrB, VopV mRNA, and VtrB + VopV mRNA displayed association trends similar to that observed for VtrB and the T3SS2 needles, where there were more VtrB puncta than corresponding VopV mRNA signals, however, these VopV mRNA puncta had a corresponding VtrB signal (Supplementary Fig. 7c). While this experiment lacked a control mRNA for the specificity of the VopV mRNA localization with VtrB, these results do demonstrate colocalization consistent with a second transertion step, where the membrane-localized master regulator of T3SS2, VtrB, initiates the transcription of T3SS2 effectors for expression and secretion through the T3SS2 apparatus assembled in its immediate vicinity.

## Discussion

Over a half-century ago, the conceptualization of transertion as a viable mechanism for the assembly of membrane proteins was proposed and was based on findings of membrane associations of the bacterial nucleoid and co-transcriptional translation in bacteria[15,16,32]. Further evidence backing the existence of the transertion phenomenon came in the form of a review article, which described the targeting and co-translational insertion of ribosome-nascent chains (a complex of a ribosome, an mRNA, and a partially synthesized polypeptide chain), into the cytoplasmic membrane via the SecYEG channel[33]. Libby et al.[18], demonstrated that the induction of membrane protein lactose permease (LacY) expression in *E. coli* was associated with a shift of the projected distances of the chromosomal *lac* locus towards the membrane. They ascertained that this observation was to be due to a decrease in the distance between the membrane and the chromosomal *lac* locus across a population of cells upon induction, as such a shift was not observed in the absence of the inducer.

While multiple studies have since explored subsets of the transertion phenomenon[18,34–37], here we have provided experimental evidence, demonstrating transertion's role in coupling a co-component signal transduction system with a membrane-bound transcription factor to mediate the localized deployment of a large virulence system in response to an external stimulus. The conservation of such co-component signal transduction system-mediated relay networks in enteric bacteria was revealed in a recent study with detailed identification of a new family of fast-evolving receptors in enteric bacteria[13]. A characteristic feature of these receptors is that they are heterodimeric in nature and, like VtrA/C, possess a lipid-binding lipocalin-like fold in their periplasmic domain for sensing lipids for the regulation of virulence systems[13]. Therefore, in addition to lending credence to the notion that membrane-bound transcription factors exist for precise spatiotemporal regulation of bacterial molecular machines, these findings set the stage for a novel area of research exploring biosynthetic components used to assemble molecules for the production of membrane-bound complexes in various enteric bacteria.

As to why transertion might be advantageous to bacteria, a number of theories have been discussed. These include ordering of heterogenous membrane proteins, organization of chromosomes, and/or efficient incorporation of proteins into the membrane[38–40]. Herein we suggest that transertion is used for the efficient assembly of a complex molecular membrane-embedded machine. For *Vibrio parahaemolyticus*, this would be its T3SS2 virulence machine.

The definition of transertion includes the concepts of colocalization and concurrence. Our data clearly illustrates the colocalization of a membrane receptor, genetic material, and a membrane-inserted product. However, only guilt by association suggests concurrence of transcription, translation, and insertion. It is possible that these processes are more dynamic than what we could capture with static pictures. Future efforts with higher resolution studies and the use of time-lapse imaging should further define this process.

## Methods

### Bacterial strains and media

All bacterial strains used in this study are listed in Supplementary Table 1. Media was procured from Fisher Scientific™ and chemical reagents from Sigma-Aldrich, unless mentioned otherwise. *E. coli* and *V. cholerae* strains were cultured at 37 °C in Miller's Luria Bertani (LB) broth with agitation. When necessary, the media was supplemented with either 50 μg/ml kanamycin, 25 μg/ml chloramphenicol or 50 μg/ml zeocin. *V. parahaemolyticus* strains were routinely cultured in LB broth supplemented with 3% (w/v) NaCl (MLB) at 30 °C, unless noted differently, and when required were supplemented with either 250 μg/ml kanamycin or 25 μg/ml chloramphenicol.

### Plasmid Construction

For extrachromosomal expression of gene fragments, the pLAU53 vector backbone was used. Firstly, the vector was PCR-linearized with primers Lf/pLAU53_nobla_KpnI and Ri/pLAU53_nobla_SpeI to remove the AmpR cassette. Then, the KanR cassette was PCR amplified from the pET28a vector (F/pET28a_KmR_KpnI & R/pET28a_KmR_SpeI), restriction digested with KpnI/SpeI and ligated into the linearized vector that had been similarly digested to give pLAU(Kan). SS2A5 was constructed by amplifying the *ygfp-parBMT1* gene fragment from TND1379 chromosomal DNA using primers F/ygfp-parBMT1_NheI and R/parBMT1_HindIII, NheI/HindIII digesting both the fragment as well as the pLAU(Kan) vector, and ligating the resultant fragment into the linearized vector backbone. For SS1I9, the synthesized 1.42 Mbp-*parSMT1* gene fragment was digested out of SS1I3 using enzymes SacI/SalI and ligated into the linearized pDM4 backbone (SacI/SalI). To clone the 0.458 Mbp-*parSP1* fragment into pDM4, primers KK23/KK45 and KK28/KK46 were first used to PCR amplify ~1 Kbp regions flanking the chromosomal site of insertion and included the *parSP1* sequence. These two fragments were then integrated into the PCR-linearized pDM4 backbone (KK21/KK22) using Gibson Assembly to make SC300.

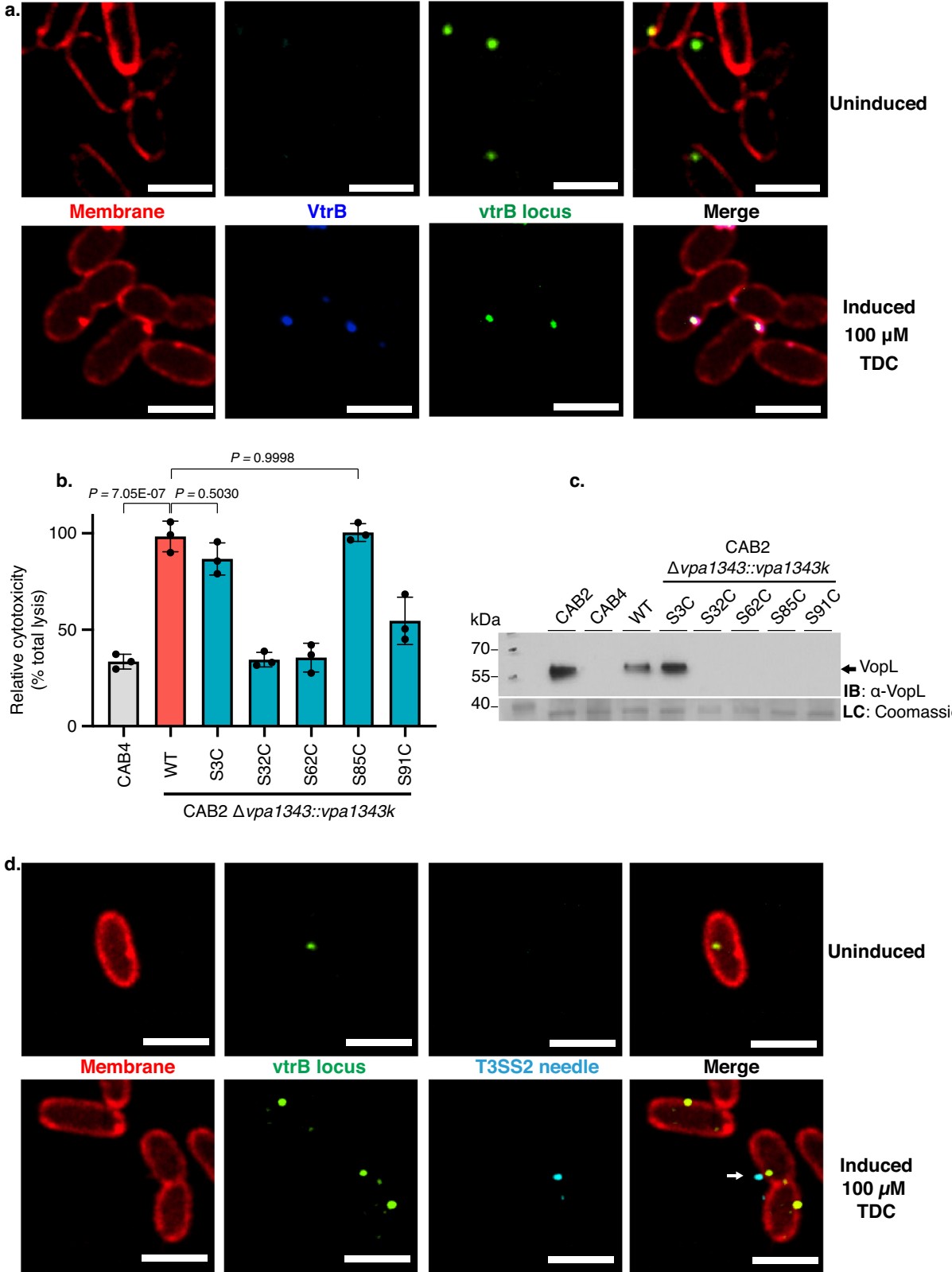

NG56 was similarly constructed by inserting P$_{vpa1348}$ (KK61/KK62) from CAB2 chromosomal into the PCR-linearized SC300 (KK59/KK60). ECKK7 [*cfp-parBP1-ygfp-parBMT1* from TND1379 (KK15/KK16) into pLAU(Kan) (KK13/KK14)], SC299 [~±1 Kbp *vpa1348* from CAB2 (KK52/KK57 and KK55/KK56) and *msgfp* from pC014-*lwcas13a-msfgfp* (KK53/KK54) into pDM4 (KK49/KK58)], and ECKK12 [*mcp* (KK68/KK69) from pDZ274 and *ecfp* (KK70/KK71) from pLAU(Kan) into pLAU(Kan) (KK66/KK67)] were constructed using a similar Gibson Assembly-based strategy. For ECKK11, ~±1 Kbp *vpa1357* from CAB2 (KK72/KK40 and KK43/KK73) and *ms2SL* from pDZ415 (KK41/KK42) were restriction digested with BglII/XhoI and inserted into the similarly digested pDM4 backbone. To generate the constructs for Cys-labeling Vpa1343, *vpa1343* including 3 Kbp upstream and downstream flanking regions were first amplified using primers F/pUC18-1343_BamHI and R/pUC18-

**Fig. 3 | VtrB is membrane inserted at the site of vtrB locus capture in the inner membrane where it facilitates assembly of the T3SS2 apparatus. a** Confocal micrographs showing expression and colocalization of VtrB (false-colored blue) with the *vtrB* genomic locus (false-colored green) at the membrane in *V. parahaemolyticus* SC295 cells cultured in inducing (100 μM TDC) vs. non-inducing conditions. **b** Bar graph depicting relative cytotoxicity of *V. parahaemolyticus* CAB2 (WT), CAB4 (T3SS2⁻) and T3SS2 needle mutants (*vpa1343S3C, vpa1343S32C, vpa1343S62C, vpa1343S85C,* and *vpa1343S91C*) in HeLa cells. Graph bars represent mean % lysis values ± SD and black dots represent individual data points for *n* = 3 technical replicates with 2 biological replicates per experiment. Statistical significance was calculated with one-way ANOVA with Tukey's multiple comparison tests and *P* values are indicated (ⁿˢ*P* > 0.05, **P* ≤ 0.05, ***P* ≤ 0.01, ****P* ≤ 0.001, *****P* ≤ 0.0001) **c** Immunoblot (IB) for the T2SS2-associated VopL effector, secreted by the strains mentioned above under inducing conditions (0.05% bile salt). Blot was stained with anti (α)-VopL (1:1000) rabbit primary antibodies and anti (α)-rabbit HRP-conjugated (1:10,000) secondaries. Identically loaded, Coomassie-stained gel was used as the loading control (LC). **d** Confocal micrographs of *V. parahaemolyticus* VPKK6 (*vpa1343S3C*) cells cultured in inducing (100 μM TDC) and non-inducing conditions, showing colocalization of T3SS2 needles (false-colored cyan) with the membrane-captured *vtrB* locus (green) at the membrane (red). White arrow points to T3SS2 needles. **a, d** Scale bar = 2 μm. **a–d** Two independent biological repetitions were performed. Source data are provided as a Source Data file.

1343_EcoRI, digested with BamHI/EcoRI and ligated into the pUC18 backbone. To aid in PCR screening of subsequent needle mutants, a KpnI restriction site was introduced into *vpa1343* at T153 using site-directed mutagenesis (F/sdm1343_KpnI and R/sdm1343_KpnI). The plasmid was digested with BglII and the resultant 1.3 Kbp *vpa1343-kpnI* was cloned into the pDM4 backbone (SS1G7). Successive Vpa1343 Cys-labeling constructs were made using site-directed mutagenesis with their corresponding primers detailed in Supplementary Table. 1.

## Strain construction and allelic exchange

Both pLAU(Kan) and pDM4-based vectors were inserted into their corresponding recipient *V. parahaemolyticus* strains via triparental conjugation facilitated by *E. coli* DH5α (pRK2043). The transconjugants were selected on minimal marine medium (MMM) agar plates containing either 250 μg/ml kanamycin or 25 μg/ml chloramphenicol for pLAU(Kan) or pDM4-derived constructs, respectively, and confirmed by PCR. Allelic exchange for in-frame deletions/additions/substitutions in *V. parahaemolyticus* strains using pDM4-based constructs was carried out by subsequent growth on MMM agar plates supplemented with 15% (w/v) sucrose for counterselection. The resultant mutants were verified by PCR and sequencing.

## T3SS2 expression and effector secretion assay

*V. parahaemolyticus* strains were grown overnight in MLB at 30 °C and were diluted the following day to OD₆₀₀ = 0.3 in fresh media. T3SS2 expression was induced by supplementing the media with 0.05% bile salts and incubating the cultures for 3 h at 37 °C[22]. Equal volumes of bacterial cultures, normalized to an OD₆₀₀ = 0.5 were centrifuged at 4000 × *g* for 10 mins. The pellet/expression fraction was resuspended in 2× Laemmli buffer. The supernatant/secretion fractions were filtered through a 0.22 μM filter and precipitated using 150 μg/ml deoxycholate and overnight treatment with 7.5% (v/v) trichloroacetic acid at 4 °C. The following day, the precipitated proteins were collected by centrifuging at 16,000 × *g* for 15 mins, washing the pellets twice in acetone and resuspending them in 2× Laemmli buffer. Western blot analysis was used to detect expression and secretion levels. Uncropped western blots and loading control gels are provided in the Source Data file.

## Lactate dehydrogenase (LDH) cytotoxicity assay

HeLa cells were plated at 7 × 10⁴ cells per well in a 24-well tissue culture plate in triplicate and were allowed to grow for 16–18 h. Bacterial cultures were induced for T3SS2 expression as described previously by growing the cells for 90 min at 37 °C. The induced *V. parahaemolyticus* cultures were added to the HeLa cells at an MOI of 10 and the infection was synchronized by immediately spinning down the plate at 1000 × *g* for 5 min. 100 μg/ml gentamycin was added to the HeLa cells 2 h post-infection. 200 μl of the spent media from each well was collected 7 h after gentamycin treatment and centrifuged in a 96-well plate at 1000 × *g* for 5 min. 100 μl of the resultant supernatant was assayed for host cell lysis using the cytotoxicity detection kit (Takara Bio) as per the instructions of the manufacturer. As a positive control, HeLa cells from the uninfected control well were treated with 1% Triton X-100 for 10 min at the end of the gentamycin treatment. HeLa cell lysis by the *V. parahaemolyticus* strains was expressed as a % of cell lysis induced by Triton treatment.

## Fluorescence labeling, widefield, and super-resolution microscopy

T3SS2 expression in *V. parahaemolyticus* strains was initiated in a manner similar to that mentioned above, using 100 μM taurodeoxycholate (TDC) as the inducing bile salt and incubating at 37 °C for 25 min. When needed, the media was additionally supplemented with 0.02% arabinose to express the ParB fusion protein of the genomic locus-tagging system. Nile Red (1 μg/ml) and Alexa Fluor™ 647 C₂ Maleimide (10 μM) were added to the cultures and incubated for a further 20 mins at 37 °C to stain the membrane and T3SS2 needles, respectively. The bacterial cells were collected by centrifugation at 6000 × *g* and fixed at room temperature for 10 min with 3.2% paraformaldehyde. The cell pellets were washed thrice with 1× PBS and were either imaged immediately or stored at 4 °C for imaging the following day. 0.5 μl of the cells were spotted on 1% agarose pads mounted onto glass slides and were allowed to dry prior to adding the coverslip. Super-resolution images were acquired as z-stacks using the Olympus Spin-SR Spinning Disk Confocal Microscope System at ×320 magnification (×100 oil objective coupled with the 3.2× super-resolution module). As mentioned, widefield images were acquired using the same imaging system at ×100 magnification to ascertain global patterns of fluorophore expression and colocalization.

## Immunogold labeling and transmission electron microscopy

*V. parahaemolyticus* strains were cultured in MLB at 37 °C for 45 mins without or with 100 μM TDC to induce T3SS2 expression. Cells were collected at 4000 × *g* for 10 mins, washed in 0.2 M phosphate buffer, and then fixed with 8% paraformaldehyde in 0.2 M phosphate buffer for 10 minutes at room temperature. After an additional wash with 0.2 M phosphate buffer, the cell pellets were resuspended in 0.1 M phosphate buffer with 4% paraformaldehyde. The samples were subsequently embedded in LR White resin, thin-sectioned, mounted on nickel-coated grids, and stained with the appropriate concentration of primary and nano-gold-conjugated secondary antibodies for transmission electron microscopy. All imaging was done on the JEOL 1400 Plus Transmission Electron Microscope equipped with a BIOSPR CCD camera at the University of Texas Southwestern Medical Center EM core facility.

## Image processing and data acquisition

All images were processed using the accompanying cellSens Dimension software. Constrained iterative deconvolution (cellSens TruSight 3D Deconvolution module) optimized for super-resolution microscopy (20 iterations) was performed to clean up image noise. The resultant images were used to calculate linear fluorescence intensity profiles (cellSens Count & Measure module). Constrained iterative deconvolution optimized for widefield microscopy (5 iterations) was performed for noise reduction of widefield images.

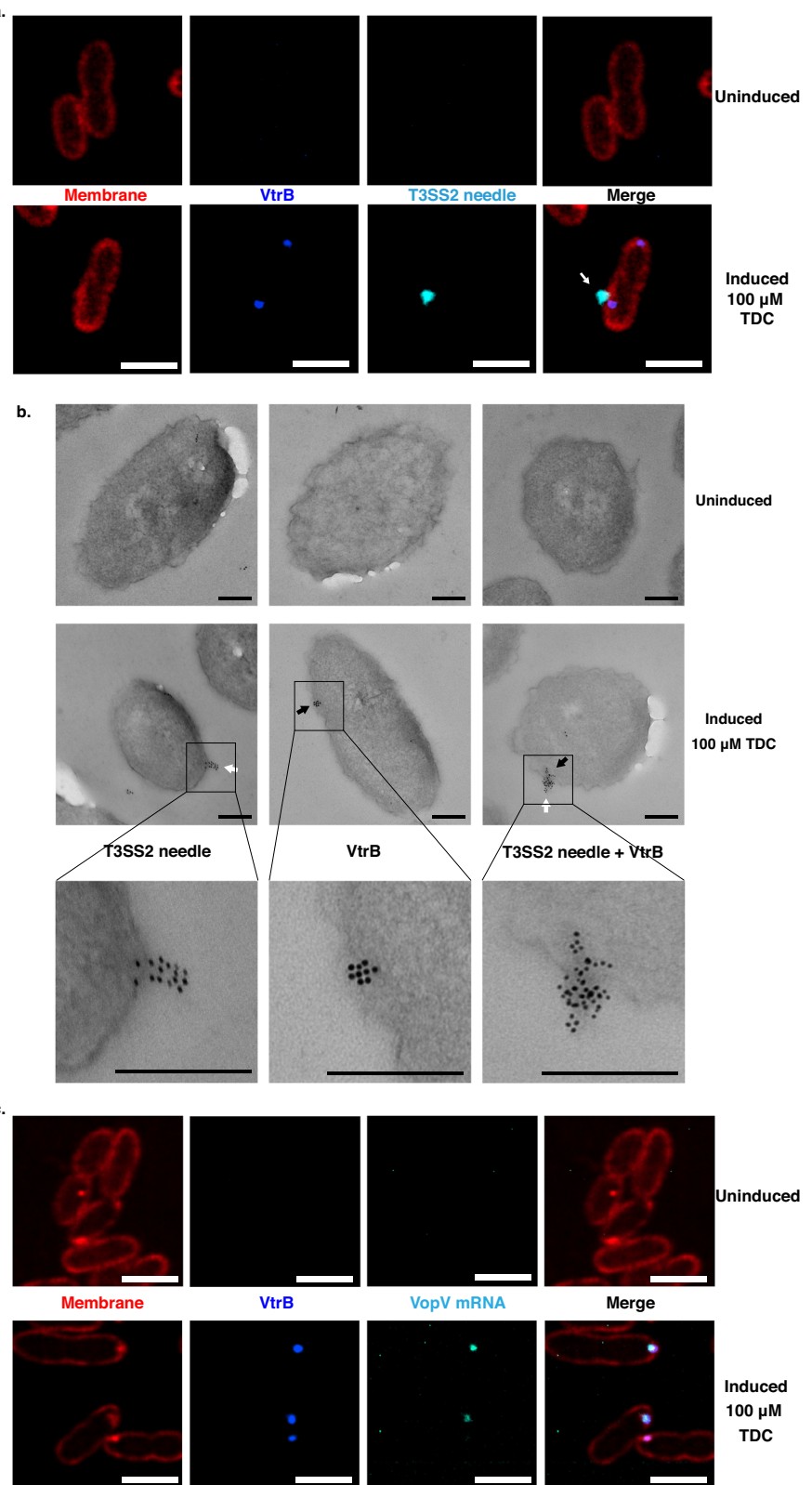

**Fig. 4 | T3SS2 effectors are transcribed in the immediate vicinity of VtrB and the T3SS2 apparatus. a** Confocal micrographs of *V. parahaemolyticus* SC288 cells cultured in inducing (100 μM TDC) and non-inducing conditions, displaying colocalization of VtrB (false-colored blue) with T3SS2 needles (false-colored cyan) at the membrane (red). White arrow points to T3SS2 needle. **b** Transmission electron micrographs of VPKK12 cells grown in either non-inducing or inducing (100 μM TDC) conditions. Thin-sectioned samples were stained with anti-GFP (1:1500) mouse and anti-Vpa1343 (1:500) rabbit primary antibodies, and counterstained with anti-mouse 12 nm gold-conjugated (1:1000) and anti-rabbit 5 nm gold-conjugated (1:1000) secondary antibodies. Black arrows point to VtrB clusters while the white arrows point to the T3SS2 needle/s. Scale bar = 200 nm. **b** Lower panel. **c** Confocal micrographs corresponding to *V. parahaemolyticus* VPKK19 cells displaying membrane colocalization of VtrB (false-colored blue) and MS2-ECFP tagged VopV mRNA (cyan) in 100 μM TDC induced cells vs. the uninduced control. **a**, **c**. Scale bar = 2 μm. **a–c** Two independent biological repetitions were performed. Source data are provided as a Source Data file.

## Statistics and reproducibility

All data are presented as their mean values ± standard deviation from two or more independent experiments, each conducted in triplicate, unless mentioned otherwise. Plotting of graphs and statistical analyses of data was performed using Prism 9 version 9.5.0. One-way ANOVA with Tukey's multiple comparison tests were used to perform statistical analyses, unless mentioned otherwise, with a $p$ value <0.05 being considered as significant. Pixel intensity correlations and colocalizations coefficient calculations (Pearson's correlation coefficients "R") were done using Coloc 2 version 3.0.5. Scatter plots of pixel intensities were plotted in ScatterJn[41] version 1.0. Both the Coloc 2 and ScatterJn modules were run in ImageJ2 version 2.9.0 (build 133148d777). Nonlinear regression analyses using the Gaussian lognormal equation were performed on the membrane proximity frequency distributions of fluorescent genomic loci. The goodness of fit of the regression curves has been denoted by $R^2$ values.

## Reporting summary

Further information on research design is available in the Nature Portfolio Reporting Summary linked to this article.

## Data availability

A reporting summary for this article is available as Supplementary Information file. The main data supporting the findings of this study are available within the article and its Supplementary Figures. The source data underlying Figs. 2–4, Supplementary Figs. 1–3 and Supplementary Figs. 6–7 are provided as a Source Data file. Specific data $P$ values are also included within the Source Data file. Additional details on datasets such as raw micrographs and protocols that support the findings of this study are available from the corresponding author upon request. Source data are provided with this paper.

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

## Acknowledgements

We thank the Orth lab members for their discussions and editing. We also thank Katherine Luby-Phelps and Phoebe Doss at the UT Southwestern Electron Microscopy Core Facility for their assistance with TEM sample preparation and use of the JEOL 1400 Plus system. The study was funded by the Welch Foundation grant I-1561, Once Upon a Time… Foundation, National Institutes of Health Grants R35 GM 134945 (to K.O.), R35 GM128674 (to A.B.D.), and National Institutes of Health Shared Instrumentation Grant 1S10OD021685-01A1 to K.L.P. K.O. is a W.W. Caruth, Jr. Biomedical Scholar with an Earl A. Forsythe Chair in Biomedical Science.

## Author contributions

K.G.K., S.C., S.D.S., and K.O. designed the experiments; K.G.K., S.C., S.D.S., and N.G. conducted experiments; K.G.K. performed microscopy; K.G.K. S.C. S.D.S., J.J., and N.G. performed cloning and strain preparation; K.G.K. and K.O. wrote the manuscript with input from all authors.

## Competing interests

The authors declare no competing interests.
