## [Peer Review File · Nature Communications]

Membrane-localized expression, production and assembly of
Vibrio parahaemolyticus T3SS2 provides evidence for
transertionEditorial Note: This manuscript has been previously reviewed at another journal that is not operating a transparent peer review scheme. This document only contains reviewer comments and rebuttal letters for versions considered at *Nature Communications*.

Reviewer #1 (Remarks to the Author):

The authors have made great progress to increase the number of measured events, so that quantification is more convincing. I also appreciated that the authors down-toned their claims.

There are a number of issues that still require more clarity from my viewpoint: The authors state that they use super-resolution, and they use a commercial system for that. To my knowledge, the used system works through SIM illumination and can indeed be powerful. It should be said though that the system has a “theoretical” capacity to provide resolution of 100nm in xy and 400nm in z (these numbers are provided by the manufacturer, I guess). Such claims are often made by the companies, however when one tests the resolution experimentally, one realizes that the resolution limit is different from the maximal one mentioned by company. This is often due to the specific sample that is investigated, and also due to how much “care” is taken when acquiring the images. I suggest that the authors include an experimental assessment of the actual resolution under the conditions used for their study. This can be easily done by an experimental determination of the PSF- This info could really help to see how robust the data is.

I appreciate the addition of the large field widefield microscopy data, nevertheless, the included data in fig 2 and 3 is still puzzling, and I was not convinced by the explanation of the authors (spots being potentially out of focus). Even if the measured event is out of focus, there should be some signal as the microscope used by the authors does not block entirely the signals that are a bit out of focus but still close (which will be the case for the bacteria). If left as it is it will remain puzzling. Alternatively, the authors could simply show some data with a z-stack, and then the signals should become visible for each measured bacterium.

I think the EM data is really difficult to interpret and as mentioned before, I have some problems with the quality of it. For example, I can see immune-gold labeling within the bacterium at the picture on the upper left side- does this mean that there is a T3SS within the bacterium?. Is the data representative? Can the authors show quantification of signals from more than a single cell? I think this would be really important.

Reviewer #2 (Remarks to the Author):

I am happy with the revised manuscript and the authors' responses to reviewer comments. I believe the manuscript is now ready for publication.

We thank the reviewers for their constructive criticisms and suggestions.

At the outset, we want to make clear that in the previously submitted manuscript, all the imaging was done using the super-resolution module with magnification of 320x (~100nm in xy and ~400nm in z). Constrained iterative deconvolution (cellSens TruSight 3D Deconvolution module) optimized for super-resolution microscopy (20 iterations) was additionally used to minimize signal noise. To the best of our knowledge, there are no automated tools to perform high throughput analyses based on our quantification strategies to measure membrane proximity of genomic loci. Therefore, this analysis must be performed manually.

In addition, we base our understanding of transertion on the definition published Libby, E. A., et al. (2012), “It has been hypothesized that for membrane proteins, transcription, translation, and insertion into the membrane are concurrent—a process termed transertion—and therefore lead to membrane localization of the encoding genes”. To the best of our knowledge, parts of this hypothesis have been supported by studies (we have cited these publications) but, until now, no one study has described this process in whole.

Below, we provide point-by-point responses for each reviewer.

REVIEWER COMMENTS

Reviewer #1 (Remarks to the Author):

The authors have made great progress to increase the number of measured events, so that quantification is more convincing. I also appreciated that the authors down-toned their claims.

There are a number of issues that still require more clarity from my viewpoint: The authors state that they use super-resolution, and they use a commercial system for that. To my knowledge, the used system works through SIM illumination and can indeed be powerful. It should be said though that the system has a “theoretical” capacity to provide resolution of 100nm in xy and 400nm in z (these numbers are provided by the manufacturer, I guess). Such claims are often made by the companies, however when one tests the resolution experimentally, one realizes that the resolution limit is different from the maximal one mentioned by company. This is often due to the specific sample that is investigated, and also due to how much “care” is taken when acquiring the images. I suggest that the authors include an experimental assessment of the actual resolution under the conditions used for their study. This can be easily done by an experimental determination of the PSF- This info could really help to see how robust the data is.

While we agree that the theoretical resolution provided by the microscope manufacturer is almost always different from the experimental resolution, experimentally determining the point spread function for every experimental condition is beyond the scope of this manuscript. Our data, with the size of the bacteria and 3D resolution is all consistent with the theoretical capacity of this microscope. To further address these concerns, we have included z-stack images (new Fig. S4 and S5) corresponding to samples shown in Fig. 3d to provide more clarity about the z-plane resolution of our images.

I appreciate the addition of the large field widefield microscopy data, nevertheless, the included data in fig 2 and 3 is still puzzling, and I was not convinced by the explanation of the authors (spots being potentially out of focus). Even if the measured event is out of focus, there should be some signal as the microscope used by the authors does not block entirely the signals that are a bit out of focus but still close (which will be the case for the bacteria). If left as it is it will remain puzzling. Alternatively, the authors could simply show some data with a z-stack, and then the signals should become visible for each measured bacterium.

As per the reviewer's recommendation, we have included z-stack of the super-resolution images in the supplementary information (Fig. S4 and S5) corresponding to the cells imaged for Fig. 3d. From these we can see that depending on the focal plane, it is possible that the fluorophores would not be visible in all bacteria even if they did express them. We have therefore also included widefield images in Fig. S3, S6 and S7.

I think the EM data is really difficult to interpret and as mentioned before, I have some problems with the quality of it. For example, I can see immune-gold labeling within the bacterium at the picture on the upper left side- does this mean that there is a T3SS within the bacterium?. Is the data representative? Can the authors show quantification of signals from more than a single cell? I think this would be really important.

Despite using multiple dilutions of our primary and secondary antibodies that were either co-applied or applied in tandem, we were unable to get images free of background noise. Also, we believe that the reason we do not see T3SS2 needles corresponding to every membrane-localized VtrB puncta, despite using a gentler gravity sedimentation method of cell collection, is due to their inherent fragility. The purpose of including the EM images was not to provide more quantification, which is done elsewhere in the manuscript using widefield imaging, but rather to support our hypothesis that the T3SS2 needles colocalize with the membrane-bound VtrB proteins when induced with bile salts.

Reviewer #2 (Remarks to the Author):

I am happy with the revised manuscript and the authors' responses to reviewer comments. I believe the manuscript is now ready for publication.

We thank this reviewer for this positive support.